# Effect of High Pressure/Heating Combination on the Structure and Texture of Chinese Traditional Pig Trotter Stewed with Soy Sauce

**DOI:** 10.3390/foods11152248

**Published:** 2022-07-28

**Authors:** Ying Wang, Yanan Sheng, Yuemei Zhang, Fang Geng, Jinxuan Cao

**Affiliations:** 1Department of Food Science and Technology, School of Food and Health, Beijing Technology and Business University, Beijing 100048, China; wangying4@nbu.edu.cn (Y.W.); zhangyuemei@btbu.edu.cn (Y.Z.); 2Department of Food Science and Technology, Ningbo University, Ningbo 315211, China; shengyanan@njau.edu.cn; 3Meat Processing Key Laboratory of Sichuan Province, School of Food and Biologicalengineering, Chengdu University, Chengdu 610106, China; gengfang@cdu.edu.cn

**Keywords:** Chinese traditional pig trotter with soy sauce, cross-links, glycosaminoglycans levels, collagen fibers, the secondary structures

## Abstract

In order to clarify the effect of a high pressure/heating combination on the texture of Chinese traditional pig trotter with soy sauce, textural parameters (springiness, chewiness, hardness, and gumminess), the secondary structures, cross-links, decorin (DCN), glycosaminoglycans (GAGs) levels, and the histochemical morphology of collagen fibers under different treatments (0.1 MPa, 150 MPa, 300 MPa, 0.1 MPa + 50 °C, 150 MPa + 50 °C, and 300 MPa + 50 °C) were assessed. At room temperature, the 150 and 300 MPa treatments increased the hardness and chewiness of the pig trotter with weak denaturation of collagen proteins compared with the control group. Textural parameters were improved at 300 MPa + 50 °C, accompanied by an ultrastructural collapse of collagen fibers, the reduction in cross-links, DCN and GAGs levels, and unfolded triple-helix structure. We concluded that the positive effects on the textural parameters of pig trotters by a combination of treatments could be attributed to the collapse of collagen structure.

## 1. Introduction

Pig trotter, marinated and stewed with soy sauce, is a popular food commodity in Asia, particularly in China and Korea, which is usually heated at about 100 °C for an extended period to improve its characteristic flavor, color, and texture [1,2,3]. The main edible tissues of pig trotters are skin and tendons, which are primarily composed of type-I collagen [4]. Collagen is the main component of collagen fiber. The triple helix of type-I collagen molecules is composed of polypeptide chains with a repetitious G-X-Y sequence, where G, X, and Y usually correspond to glycine, proline, and hydroxyproline, respectively. The assembly of type-I collagen molecules leads to the formation of microfibrils of a few nanometers in diameter, which can further aggregate to form collagen fibrils with diameters of approximately a few hundred nanometers. Fibrils are not usually found alone but rather bundle to form collagen fibers that are microscale structures [4].

Although some Western and European customers classify them as “abnormal food”, Chinese people believe that the intake of collagen peptides through eating pig trotter has the potential benefits to promote human health and prevent wrinkles. In Chinese markets, the price of pig trotter stewed with soy sauce has risen to about CNY 130/kg, while that of “normal” meat products is about CNY 55/kg. Stewing is a common cooking method, which can effectively improve the flavor and texture of meat products [5]. This has been demonstrated to decrease the hardness, springiness, cohesiveness, and chewiness in subcutaneous fat from the hind legs and back, which could be caused by the melting and gelation of the connective tissue [6]. In a previous study, we (2012) reported that the prolonged stewing times of 60 and 120 min contributed to the improvement of texture by a reduction in cross-linkage, decorin (DCN) and glycosaminoglycan (GAG) levels of collagen fibers, and the unfolding the triple-helix structure of collagen [7]. However, a long time and great effort are required to obtain the soft texture of pig trotter.

High-pressure treatment, a non-thermal technology, is being widely used in the meat industry to promote protein denaturation, shorten the processing time of the product, and improve its texture [8]. Over the past few decades, the combined treatment of high pressure and heating, as an advanced technology, has been widely developed in muscle food processing to accelerate changes in myofibrillar proteins [8]. The high pressure can not only lead to the loss of the structural continuity of a sarcomere through the break of A- and I- filament spread over the sarcomere, but also cause the disappearance of the M-line and thickening of the Z-line [9]. It has been reported that the denaturation of myosin occurred with the increase in temperature [10]. However, the effects of combined high pressure and heating treatments on the texture and structure of pig trotter, which was composed of collagen tissue, has not been reported yet.

In the present work, in order to shorten the processing time and improve the texture of pig trotter, the combined treatment of high pressure and heating was employed to study various indicators of pig trotters. In this study, we characterized DCN and GAGs content, the secondary structure and the histochemical morphology of collagen fibers, the degree of cross-linking, and the textural profile of pig trotter.

## 2. Materials and Methods

### 2.1. Procedures for Pig Trotter Stewed with Soy Sauce

A total of 60 pieces of raw pig trotter with an average weight of approximately 0.23–0.31 kg from Duroc × Landrance cross-breed pigs were purchased from a local processing plant. Then, they were washed after the toes were cut off. The rectangular pig skin (about 2 cm × 1 cm × 3.0 mm) was taken from the pig trotters and was sampled after removing subcutaneous fat. The pig skin samples were divided into 6 groups with 10 samples in each group. Each rectangular pig skin was put in an individual polythene bag (oxygen permeability of 40–50 cm^3^/m^2^/day at 20 °C) and vacuum packed immediately. The vacuum-packed samples were divided into 6 groups with 10 repeats for each treatment. The treatments used (all in the same day) were: (a) the control group (0.1 MPa) was non-pressurized (NP) at room temperatures (RT); (b) the 150 MPa group was treated under 150 MPa in a high-pressure equipment (Stansted Fluid Power Ltd., Harlow, England) at RT for 15 min; (c) the 300 MPa group was treated under 300 MPa at RT for 15 min; (d) the 0.1 MPa + 50 °C group was treated under NP at 50 °C for 15 min; (e) the 150 MPa + 50 °C group was treated under 150 MPa at 50 °C for 15 min; and (f) the 300 MPa + 50 °C group was treated under 300 MPa at 50 °C for 15 min. The internal volume of the vessel was 1 L. Pressure treatment time excluded pressure come up and releasing time. The rate of pressure increase was about 100 MPa/min; releasing time was just 10 s to minimize adiabatic heating. After treatment, the vacuum-packed bags were removed, and pig skins were stored at 4 °C until further analysis.

### 2.2. Texture Profile Analysis (TPA) Measurements

Before the textural analysis, rectangular pig skin (about 2 cm × 1 cm × 3.0 mm) was taken from pig trotters for convenience after visible hair was removed. All samples were measured by a texture analyzer (Model, TA. XT; Make, Stable micro systems, Godalming, UK), which was equipped with a load cell of 1 kg and a cylindrical probe (P/5). In each case, intact tissue was placed in the center of the object stage and compressed to 40% of its original height [11]. The probe always returned to trigger point for a rest period of 3 s before the second cycle started. After the second cycle, the probe returned to its initial position. The pre-test speeds, test speeds and post-test speeds were 2 mm/s, 0.5 mm/s, and 2 mm/s, respectively.

The measured parameters were springiness, hardness (g), chewiness, and gumminess. The data obtained from the TPA curve were used for the calculation of textural parameters [12]. Hardness is expressed as the maximum force for the first compression. Springiness is calculated as the ratio of time from the start of the second area up to second probe reversal over the time between the start of first area and first probe reversal. Chewiness is calculated as hardness × cohesiveness × springiness. Gumminess is presented as hardness × cohesiveness [13]. All measurements were taken at room temperature and repeated 10 times.

### 2.3. Histochemical Morphology

The histology analysis was conducted as described previously by Gurniak et al. [14] with some minor modification. For paraffin sections, pig skin tissues (about 0.5 cm × 0.5 cm × 0.5 cm) from pig trotters, which were fixed by 4% paraformaldehyde, were dehydrated using JJ-12J Dehydrator (Junjie Electronics Co., Ltd., Wuhan, Hubei, China) by the following sequence: a graded series (75% for 4 h, 85% for 2 h, 90% for 2 h, 95% for 1 h, and 100% for 1 h) of ethanol, alcohol benzene (ethanol: xylene = 1:1, *v*/*v*) for 30 min, dimethylbenzene Ⅰ for 10 min, dimethylbenzene Ⅱ for 10 min, paraffin wax Ⅰ for 1 h, paraffin wax Ⅱ for 1 h and paraffin wax Ⅲ for 1 h. Then, the tissues were embedded into paraffin wax by JB-P5 Embedding machine (Junjie Electronics Co., Ltd., Wuhan, China). The embedded samples were trimmed to 3 μm sections parallel to the collagen fiber direction and stained with routine procedures [15]. Sections were dewaxed in xylene Ⅰ and xylene Ⅱ for 20 min, respectively, which was followed by hydration in a graded series of ethanol (100% ethanol Ⅰ for 10 min, 100% ethanol Ⅱ for 10 min, 95% for 5 min, 90% for 5 min, 80% for 5 min, and 70% for 5 min). After being rinsed in deionized water for 5 min, sections were stained with hematoxylin and eosin. The resultant samples were dehydrated in the subsequent steps with 5 min for each step: 95% ethanol Ⅰ, 95% ethanol Ⅱ, 100% ethanol Ⅰ, 100% ethanol Ⅱ, xylene Ⅰ, and xylene Ⅱ. Finally, the samples were sealed in neutral gums. The histomorphological appearance of collagen fibers was observed by a Nikon Eclipse Ti-SR inverted fluorescence microscope (Olympus, Nikon, Tokyo, Japan).

### 2.4. Raman Spectroscopy

The collagen fiber of the skin tissue from pig trotters was measured using a Raman analyzer (Renishaw in Via Reflex Raman spectrometer; Renishaw, Gloucestershire, UK) according to Alix et al. [16]. The ground and freeze-dried skin tissue samples were placed on microscope slides. An argon-ion laser that emitted a wavelength of 785 nm was used as an excitation source. The scattered radiation was collected at 180° to the source; the typical spectra were recorded at 1 cm^−1^ resolution with 500–2100 cm^−1^ scans with 12 mW of laser power. The Phe m-ring band located near 1003 cm^−1^ was used as the internal standard for the normalization of spectra. Eventually, the relative content of the collagen secondary structures (α-helix, β-sheet, β-turn, and random coil) was calculated by Peak Fit 4.12 (Sea Solve Software Inc., San Jose, CA, USA) according to our previous methods [17].

### 2.5. Determination of Cross-Links Degree, DCN and GAGs Content

The degree of cross-links, the content of total collagen, and DCN and GAGs were measured according to previous reports [18,19]. The samples were processed as follows: 1 g of skin tissue from pig trotters was homogenized by DY89-I high-speed homogenizer (Scientz co., Ningbo, Zhejiang, China) in 10 mL of phosphate-buffered saline (0.02 M, pH = 7.2) on the ice. Subsequently, the homogenized solution was centrifuged with a refrigerated centrifuge (Hunan Xiangyi, Laboratory Instrument Development Co., Changsha, China) at 1500 g for 15 min at 4 °C. Finally, the supernatant was collected for the determination of cross-links degree, total collagen, and DCN and GAGs content with enzyme-linked immunoassay kits. A cross-links Elisa assay kit (E07P0743, Blue Gene Biotechnology Co., Ltd., Shanghai, China) with a pre-embedded mouse anti-cross-links monoclonal primary antibody, a total collagen Elisa assay kit (E07T0752, Blue Gene Biotechnology Co., Ltd., Shanghai, China) with a pre-embedded mouse anti-collagen monoclonal primary antibody, a porcine DCN Elisa assay kit (E07D0145, Blue Gene Biotechnology Co., Ltd., Shanghai, China) with a pre-embedded mouse anti-DCN monoclonal primary antibody, and a GAGs Elisa assay kit (E07G0301, Blue Gene Biotechnology Co., Ltd., Shanghai, China) with a pre-embedded mouse anti-GAGs monoclonal primary antibody and corresponding HRP conjugated rabbit anti-mouse secondary antibodies were used to determine the cross-links degree, total collagen, and DCN and GAGs content, respectively, according to the instructions of the manufacturer. The absorption was measured at 450 nm with a 96-well Plate Reader M200 (Tecan, Austria) to calculate the concentration. The results are expressed in μmol of pyridinoline cross-links per g of collagen (μmol/g), μg of DCN per g of total collagen (μg/g), and mg of GAGs per g of total collagen (mg/g).

### 2.6. Statistical Analysis

The experimental plan was designed as completely randomized. All data were expressed as means of 10 replicated samples. All data of figures and tables were presented as mean ± standard error. The effect of 6 treatments on the cross-links, GAGs, DCN and secondary structure contents of collagen and the texture parameters of pig trotters was analyzed via the one-way analysis of variance procedure. Means were compared using Duncan’s multiple range test of SAS 8.0 software (SAS Institute Inc., Cary, NC, USA). The level of statistical significance of differences was set as *p* < 0.05.

## 3. Results

### 3.1. The Textural Parameters

Figure 1 shows the effects of different conditions (0.1 MPa, 150 MPa, 300 MPa, 0.1 MPa + 50 °C, 150 MPa + 50 °C, and 300 MPa + 50 °C) on the textural parameters (springiness, chewiness, hardness, and gumminess) of pig trotters (Figure 1). The springiness levels of the pig trotters treated at 300 MPa, 150 MPa + 50 °C, and 300 MPa + 50 °C were significantly lower than that of the control group (*p* < 0.05). At room temperature, the high-pressure treatment led to a significant increase in chewiness, hardness and gumminess compared with the control group (*p* < 0.05). After treatments of combined high pressure and heating, the gumminess of the sample treated at 150 MPa + 50 °C was significantly higher than that of the control group (*p* < 0.05), whereas the gumminess of the pig trotters treated at 300 MPa + 50 °C was significantly lower than that of the control group (*p* < 0.05). The results indicate that only single high-pressure treatment is not enough to improve the texture of pig trotters. However, high-pressure combined heating treatments at 300 MPa + 50 °C have a positive impact on the texture parameters (low springiness, chewiness, hardness, and gumminess values) of pig trotters.

In our previous results, the disruption and depolymerization of collagen fibers and the dissolution of collagen proteins during stewing could contribute to the improved texture of pig trotters [7]. The high-pressure combined heating treatments could induce a disruption of collagen fibers.

### 3.2. The Ultrastructure of Collagen Fibers

The changes in collagen fibers by Van Gieson (ponceau) staining were shown in this study (Figure 2). The ultrastructure of collagen fibers for the samples treated at 0.1 MPa and 0.1 MPa + 50 °C showed a very clear and classic fiber outline and intervals among fibers, while samples treated at 150 MPa had darker collagen fibers and more obvious fiber shrinkage. The pig trotters treated at 150 MPa + 50 °C had clear fiber bands of collagen fibers, whereas the ultrastructure showed an obscure and swollen overall appearance and a significant disruption and fusion of collagen fibers. The results demonstrate that single high-pressure treatment lead to pig trotters having larger gaps between collagen fibers, which further resulted in poor meat quality. For the samples treated at 300 MPa + 50 °C, the depolymerization of collagen fibers and dissolution of collagen were observed, which apparently had a positive influence on the texture of the pig trotters.

### 3.3. The Secondary Structure of Collagen Proteins

The secondary structure of collagen in pig trotters was analyzed by Raman spectroscopy. The Raman bands of collagen I (1600–1700 cm^−1^) in collagen under different conditions were deconvoluted and curve-fitted as shown in Figure 3. Different peak regions were designated as secondary structure peaks [20,21]. The bands in the range of 1650–1665 cm^−1^ were classified as characteristic peaks of α-helices; 1670–1680 cm^−1^ and 1620–1632 cm^−1^ could be classified as the characteristic peak frequency ranges of β-sheet. The signals in the range of 1635–1639 cm^−1^ and 1680–1690 cm^−1^ were assigned to a β-turn structure, while those at 1640–1645 cm^−1^ and 1666–1670 cm^−1^ were mainly ascribed to random coil structures. The main indicators of secondary structure changes and local environmental changes in collagen were changes in the frequency and intensity of the Raman bands. The results present that the intensity of an α-helix peak for samples treated by single high-pressure and combined treatments was lower than that of the control group, whereas the intensity of the random coil peak was higher than that of the control group. The compositions of secondary structures in collagen proteins are stated clearly in Table 1. Compared with the control group, α-helix experienced a significant decrease (*p* < 0.05) while random coil witnessed a significant increase (*p* < 0.05) with increasing pressure or temperature. In addition, the content of random coil was the highest at 300 Mpa + 50 °C compared with other samples. It was concluded that the weak denaturation of collagen fibers occurred under single high-pressure treatment and treatment at 150 MPa + 50 °C, which finally led to the shrunk and hardened collagen fiber. The combined 300 MPa + 50 °C treatment not only gave rise to the intensive denaturation and complete cracking of collagen fibers, but also resulted in the dissolution of collagen and the degraded mechanical properties of collagen fibers. The final results show that although collagen was denatured under room temperature and at 50 °C, the degree of collagen denaturation was the most serious under the treatment of 300 MPa + 50 °C compared to that under other treatments. 

### 3.4. The Cross-Links, DCN, and GAGs Levels

The content changes in the cross-linking degree, DCN, and GAGs of collagen in pig trotters during different treatment condition are shown in Table 2. The consequences indicate that all the indicators of these experimental groups decreased compared with the control group. For fresh pig trotters, the levels of cross-linking degree, DCN, and GAGs were 12.43 μmol/g, 34.15 μg/g, and 1.69 mg/g, respectively. The values of cross-linking degree, DCN, and GAGs had no significant changes compared with the control (*p* > 0.05). When the pressure was 300 MPa + 50 °C, the degree of cross linking, DCN, and GAGs decreased by 15.61%, 15.46%, and 42.60% (*p* < 0.05), respectively. The results show that high-pressure combined heating treatments at 300 Mpa + 50 °C on pig trotters not only reduced levels of cross-linking degree, DCN, and GAGs significantly, but also caused the destruction and collapse of collagen fibers. However, single temperature or pressure increases had a limited impact on the content of cross-linking degree, DCN, and GAGs.

## 4. Discussion

### 4.1. Effects of Different Treatments on the Textural Parameters

High-pressure treatment can change the physical and rheological properties of proteins and result in important improvements in texture properties such as the springiness, hardness, and gumminess of samples [22]. The increase in chewiness, hardness, and gumminess could be explained by the shrinkage of collagen fibers. It was very similar to the reports of Ma and Ledward [23]. They found that high-pressure combined heating treatment used to set cod muscle gel caused some loss of rigidity, while with beef this result was not seen, and treatment at 70 °C actually caused an increase. This was due to the shrinkage of the connective tissue network in beef upon heat treatment, a phenomenon which has little effect on white fish texture. The decrease in textural parameters at 300 MPa + 50 °C means an improvement in all the texture characters. According to Ma’s report [23], it would appear that the simultaneous application of high temperature and moderate pressure (200 MPa) gives rise to a softer texture than single heating or pressure on beef muscle.

### 4.2. Effects of Different Treatment Condition on the Ultrastructure of Collagen Fibers

Related researchers have suggested that high pressure combined with cooking and salt contributes to a completely disrupted structure. The fiber outlines were barely detectable and the fiber integrity lost, which was due to the accumulation of denaturation in proteins [24]. It was also observed that the application of pressure or heat treatment alone led to the presence of low shear-force values, which were linked to the poor mechanical characteristic of these samples [25]. Our results demonstrate that the destruction and depolymerization of collagen fibers not only weakened the mechanical strength and resistance of pig trotters, but also led to a decrease in springiness, chewiness, hardness and gumminess.

### 4.3. Effects of Different Treatment Conditions on the Secondary Structure of Collagen Proteins

Hydrogen bonds were the main chemical bonds between the chains that maintained the triple-helix structure of collagen [26,27]. Some researchers reported that high-pressure treatment could not only break the balance of hydrogen bonds that stabilized the triple-helix structure of collagen, but also exert various effects on collagen and induce reversible or irreversible structural modifications, which resulted in the denaturation and aggregation of collagen [28,29,30]. As temperature increased with the 300 MPa pressure treatment, the triple-helix structure was severely damaged due to the disruption of hydrogen bonds. In our results, thermal denaturation combined with pressure treatment on collagen typically caused the disruption of collagen fibers and dissolution of collagen proteins and induced the unfolding of the triple helix to a random coil, which was consistent with other reports [31].

### 4.4. Effects of Different Treatment Conditions on Cross-Links, DCN and GAGs Levels

Cross-linking played an important role in maintaining the stability and mechanical properties of collagen networks. The relevant literature indicated that the integrity of collagen affected the mechanical properties of the collagen network, especially when collagen denaturation occurred [32]. Avery et al. also reported that mature cross-linking affected the texture of meat [33]. Lepeit et al. studied the relationship between total cross-linking and elastic modulus in the connective tissue of cooked meat, and believed that the total amount of cross-linking degree was related to the quality of the meat [34]. The results of this experiment are consistent with that of Gekko and Koga [35] and Heremans [36]. Collagen was less affected by low pressure, which was owing to the fact that hydrogen bonds were not sensitive to low pressure. Researchers showed that high-pressure treatments above 150 MPa induced protein denaturation by disrupting the equilibrium of non-covalent interactions that stabilized the natural conformation of collagen [37]. According to McCormick [38], the reduction in mature cross-linking meant a reduction in the “bridge” between collagen molecules. Therefore, the reduction in cross-linking could contribute to the destruction of the thermal stability of collagen [39]. The triple-helical hydrogen bond and covalent cross-linking were attenuated with the unfolding of the triple helix structure of collagen under high-pressure combined heating treatments [40], which affected the texture of the pig trotters.

DCN played a significant role in the assembly process of collagen and was closely related to the normal structural function of tissues [41,42]. According to the theory of Nishimura et al. [43], the decrease in DCN was positively correlated with the inhibition of collagen fiber formation. In addition, the reduction in DCN fully demonstrated that its modification of collagen fibers was weakened, and GAGs lost attachment sites [44,45,46] which made the collagen fiber structure unstable. Therefore, a decrease in the content of DCN and GAGs indicated the destruction or dissolution of collagen fibers. Pedersen et al. [47] showed that the content of core proteoglycan was also high in meat with higher hardness. The experimental results further explain that the degradation of cross-linking degree, DCN and GAGs could ultimately improve the texture of pig trotters.

## 5. Conclusions

The single high-pressure treatment failed to improve the hardness and chewiness of pig trotters, with a weak denaturation of collagen proteins. The high pressure/heating combination at 300 MPa + 50 °C improved the textural parameters of the pig trotter by inducing the ultrastructural collapse of collagen fibers, reducing cross-links, DCN and GAGs levels, and unfolding the triple-helix structure of collagen in pig trotters.

## Figures and Tables

**Figure 1 foods-11-02248-f001:**
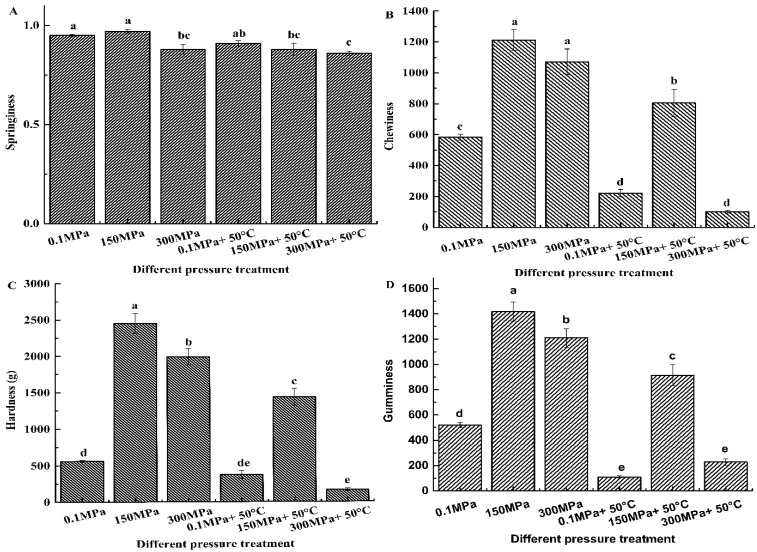
The effect of different treatment conditions (0.1 MPa, 150 MPa, 300 MPa, 0.1 MPa + 50 °C, 150 MPa + 50 °C, 300 MPa + 50 °C) on the texture (springiness, hardness (g), chewiness and gumminess). All values are means ± SE. ^abcde^ The different letters indicated the significant difference.

**Figure 2 foods-11-02248-f002:**
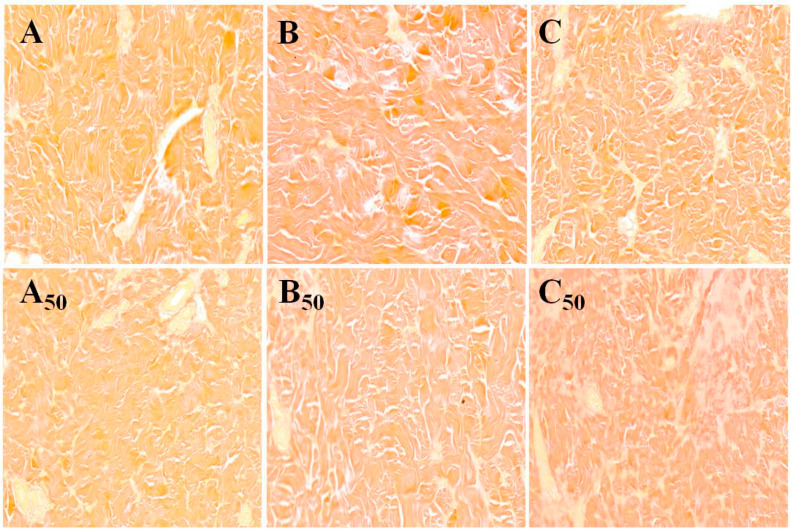
The effect of different treatment conditions (0.1 MPa, 150 MPa, 300 MPa, 0.1 MPa + 50 °C, 150 MPa + 50 °C, 300 MPa + 50 °C) on the ultrastructure of collagen fibers. A, B and C represents magnification × 200 of 0.1 MPa, 150 MPa, and 300 MPa at room temperature, respectively; A_50_, B_50_ and C_50_ represents magnification × 200 of 0.1 MPa, 150 MPa, and 300 MPa + 50 °C, respectively.

**Figure 3 foods-11-02248-f003:**
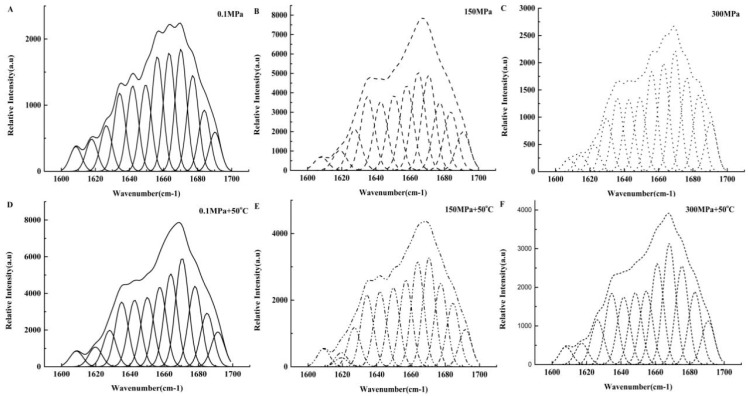
The effect of different treatment conditions (0.1 MPa, 150 MPa, 300 MPa, 0.1 MPa + 50 °C, 150 MPa + 50 °C, 300 MPa + 50 °C) on the Raman spectra of collagen.

**Table 1 foods-11-02248-t001:** The effect of different treatment conditions (0.1 MPa, 150 MPa, 300 MPa, 0.1 MPa + 50 °C, 150 MPa + 50 °C, 300 MPa + 50 °C) on the secondary structure of collagen.

Secondary Structure (%)	Different Treatment Conditions					
	0.1 MPa	150 MPa	300 MPa	0.1 MPa + 50 °C	150 MPa + 50 °C	300 MPa + 50 °C
α-helix	36.93 ± 0.58 ^a^	33.35 ± 0.82 ^b^	31.12 ± 1.21 ^bc^	34.00 ± 0.62 ^ab^	34.06 ± 0.87 ^ab^	30.06 ± 1.27 ^c^
β-sheet	33.08 ± 0.57 ^a^	31.60 ± 2.18 ^ab^	26.61 ± 2.36 ^bc^	35.25 ± 1.17 ^a^	31.16 ± 1.81 ^ab^	22.18 ± 1.39 ^c^
β-turn	20.38 ± 1.00 ^b^	24.46 ± 1.80 ^a^	23.14 ± 1.14 ^a^	22.40 ± 1.60 ^a^	22.82 ± 1.75 ^a^	23.26 ± 1.21 ^a^
Random coil	9.61 ± 0.47 ^c^	10.60 ± 1.11 ^c^	19.14 ± 2.59 ^b^	8.34 ± 0.40 ^c^	11.97 ± 1.94 ^c^	24.50 ± 1.13 ^a^

All values are means ± SE. Different letters within column and lines indicate significant differences among treatments (*p* < 0.05) (Duncan’s multiple range test).

**Table 2 foods-11-02248-t002:** The effect of different treatment conditions (0.1 MPa, 150 MPa, 300 MPa, 0.1 MPa + 50 °C, 150 MPa + 50 °C, 300 MPa + 50 °C) on the cross-links, DCN, and GAGs of collagen.

Indicators	Different Treatment Conditions					
	0.1 MPa	150 MPa	300 MPa	0.1 MPa + 50 °C	150 MPa + 50 °C	300 MPa + 50 °C
Cross-links (μmol/g collagen)	12.43 ± 0.33 ^a^	11.64 ± 0.33 ^a^	12.58 ± 0.32 ^a^	11.63 ± 0.34 ^a^	12.07 ± 0.33 ^a^	10.49 ± 0.30 ^b^
Decorin (μg/g collagen)	34.15 ± 0.89 ^a^	31.89 ± 0.81 ^a^	33.16 ± 0.86 ^a^	31.54 ± 0.73 ^a^	32.73 ± 0.85 ^a^	28.87 ± 0.89 ^b^
GAGs (mg/g collagen)	1.29 ± 0.04 ^c^	1.60 ± 0.04 ^a^	1.66 ± 0.06 ^a^	1.45 ± 0.04 ^b^	1.30 ± 0.06 ^c^	0.97 ± 0.07 ^d^

All values are means ± SE. Different letters within columns and lines indicate significant differences among treatments (*p* < 0.05) (Duncan’s multiple range test).

## Data Availability

Data is contained within the article.

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
