# Peer review of "Effect of High Pressure/Heating Combination on the Structure and Texture of Chinese Traditional Pig Trotter Stewed with Soy Sauce"

_foods, 2022, doi:10.3390/foods11152248_

Round 1

Reviewer 1 Report

What is the experimental plan? (randomized complete block design or completely randomized design). 

The experimental design should be given as a Table in the material and method section.

Were the experiments conducted at different times or was the same raw material used?

Statistical analysis should be explained clearly. 

The results of texture profile analysis are not discussed adequately.

Author Response

Comments and Suggestions for Authors

What is the experimental plan? (randomized complete block design or completely randomized design). 

Response: thanks for your suggestion! We changed it accordingly.

The experimental plan was designed as completely randomized design. All data were expressed as means of 10 replicated samples. All data of figures and tables were presented as mean ± standard error. The effect of 6 treatments on the cross-links, GAGs, DCN and secondary structure contents of collagen and the texture parameters of pig trotters was analyzed via the one-way analysis of variance procedure; means were compared using Duncan’ multiple range test of SAS 8.0 software.

The experimental design should be given as a Table in the material and method section.

Response: thanks for your suggestion! We changed it accordingly.

The treatments used (all in the same day) were: (a) The control group (0.1 MPa) was no-pressurized (NP) at room temperatures (RT); (b) the 150 MPa group was treated under 150 MPa in a high-pressure equipment (Stansted Fluid Power Ltd., Harlow, England) at RT for 15 min; (c) the 300 MPa group was treated under 300 MPa at RT for 15 min; (d) the 0.1 MPa+50 oC group was treated under NP at 50 °C for 15 min; (e) the 150 MPa + 50 oC group was treated under 150 MPa at 50 °C for 15 min; (f) the 300 MPa + 50 oC group was treated under 300 MPa at 50 °C for 15 min.

Were the experiments conducted at different times or was the same raw material used?

Response: thanks for your suggestion! We changed it accordingly.

The effect of 6 treatments on the cross-links, GAGs, DCN and secondary structure contents of collagen and the texture parameters of pig trotters was analyzed via the one-way analysis of variance procedure; means were compared using Duncan’ multiple range test of SAS 8.0 software.

Statistical analysis should be explained clearly. 

Response: thanks for your suggestion! We changed it accordingly.

The results of texture profile analysis are not discussed adequately.

Response: thanks for your suggestion! We changed it accordingly.

We developed a discussion as following:

In our previous results, the disruption and depolymerization of collagen fibers and the dissolution of collagen proteins during stewing could contribute to the improved texture of pig trotters [7]. The high pressure combined heating treatments could induce a disruption of collagen fibers.

Reviewer 2 Report

This work dealt with the effect of high pressure combined with heating treatments on the texture of Chinese traditional pig trotter.

General aspects: In its current state, the English level throughout the manuscript does not meet the journal's standard and should be improved in all sections. Before it can be considered for publication, several issues need to be addressed in this paper.

Title

The title should be improved to be clearer.

Abstract

-Lines 17-19: Please, improve this part to make it clearer.

Keywords

I suggest replacing the keywords that are already in the title of the manuscript with other words.

Introduction

Line 33: Change praline to proline

Line 37: Microscale

Line 42: 1.2-3.5 times higher or lower? Please, add this information.

Line 44: Why does stewing decrease the fragmentation of the collagen network? Wouldn't it be the opposite? Please review this information.

Line 45: Add a period after [6].

Line 50: HP technology is not new.

Line 54: Please, complete the sentence. Example: changes in the structure of myofibrillar proteins.

Material and Methods

Lines 70-71: Why did the authors only use a rectangular piece of the pig trotter and not the entire feet to apply the high pressure? This sample does not represent the product.

Lines 72-75: I do not understand why the samples were divided into 6 groups of ten samples and then into 5 groups of ten samples. Please rewrite to make this part clearer.

Line 81: der: whats does this mean?

Line 81: "the pressure of 300 MPa for 15 min". I did not understand this part of the sentence.

Line 85: Change to stored at 4 °C until further analysis.

Line 86: It was previously mentioned that 6 groups were vacuum-packed. Please clarify all information in this section. I suggest that the authors present a table or figure of the experimental design, as this section is too confusing to understand the treatments.

Line 99: Authors should not use the parameter gumminess. Chewiness only applies to solid products, whereas gumminess only applies to semi-solid products. Gumminess is mutually exclusive with chewiness since a product would not be both semi-solid and solid at the same time.

Line 104: I suggest removing this sentence since the authors are explained below how each one is calculated.

Line 106: How many measurements for each treatment?

Line 109: In this type of citation, the name of the authors should be added. Ex: by Lou et al. [14].

Line 123: "eosinThe". Please, add a space.

Line 131: In this type of citation, the name of the authors should be added. Ex: by Lou et al. [14].

Lines 146-147: The author used respectively but mentioned just one method. Please, rewrite.

Lines 158-159: Was this wavelength used to determine the four analyses? (cross-linked degree, total collagen, DCN, and GAGs)? This was not clear.

Line 163: At the beginning of the material and methods, it was mentioned that each group had 10 samples and in the statistical analysis, it was informed that the results were expressed as means of fifteen replicate samples. Please clarify.

Results

Line 171: In the Material and Methods, it was indicated that the temperature of these treatments was 25 ºC and here, the temperature was not mentioned. Please add the information of the temperature throughout the manuscript.

Line 173: Same as previously mentioned about the gumminess determination.

Line 249: "onlyreduced". Please, add a space

Line 264: In this type of citation, the name of the authors should be added. Ex: by Lou et al. [14].

Lines 307-308: Rewrite to make it clearer.  

Conclusions

The conclusion is basically presenting the results again. The conclusion is an answer to your goals, not a summary of results. I suggest the authors rewrite this section.

Line 325: "wereprocessed" Please, add a space.

Author Response

Comments and Suggestions for Authors

This work dealt with the effect of high pressure combined with heating treatments on the texture of Chinese traditional pig trotter.

General aspects: In its current state, the English level throughout the manuscript does not meet the journal's standard and should be improved in all sections. Before it can be considered for publication, several issues need to be addressed in this paper.

Title

The title should be improved to be clearer.

 Response: thanks for your suggestion! We changed it accordingly.

Abstract

-Lines 17-19: Please, improve this part to make it clearer.

  Response: thanks for your suggestion! We changed it accordingly.

Keywords

I suggest replacing the keywords that are already in the title of the manuscript with other words.

 Response: thanks for your suggestion! We changed it accordingly.

Introduction

Line 33: Change praline to proline

Response: thanks for your suggestion! We changed it accordingly.

Line 37: Microscale

Response: thanks for your suggestion! We changed it accordingly.

Line 42: 1.2-3.5 times higher or lower? Please, add this information.

Response: thanks for your suggestion! We changed it accordingly.

Line 44: Why does stewing decrease the fragmentation of the collagen network? Wouldn't it be the opposite? Please review this information.

Response: thanks for your suggestion! We made a mistake and changed it accordingly.

Line 45: Add a period after [6].

Line 50: HP technology is not new.

Response: thanks for your suggestion! We changed it accordingly.

Line 54: Please, complete the sentence. Example: changes in the structure of myofibrillar proteins.

 Response: thanks for your suggestion! We changed it accordingly.

Material and Methods

Lines 70-71: Why did the authors only use a rectangular piece of the pig trotter and not the entire feet to apply the high pressure? This sample does not represent the product.

Response: In our experimental process, the internal volume of the high-pressure equipment (Stansted Fluid Power Ltd., Harlow, England) was 1 L, but the entire pig feet was very big so that we can not put them into the vessel. According to our sensory experiment experience on pig trotter stewed with soy sauce, we find that the texture of rectangular pig skin was relative to the texture of real products. We know that it could not represent the product with a hundred percent, but we will try to improve the experiment in the future. Thanks for you comments.

Lines 72-75: I do not understand why the samples were divided into 6 groups of ten samples and then into 5 groups of ten samples. Please rewrite to make this part clearer.

Response: thanks for your suggestion! We changed it accordingly.

The treatments used (all in the same day) were: (a) The control group (0.1 MPa) was no-pressurized (NP) at room temperatures (RT); (b) the 150 MPa group was treated under 150 MPa in a high-pressure equipment (Stansted Fluid Power Ltd., Harlow, England) at RT for 15 min; (c) the 300 MPa group was treated under 300 MPa at RT for 15 min; (d) the 0.1 MPa+50 oC group was treated under NP at 50 °C for 15 min; (e) the 150 MPa + 50 oC group was treated under 150 MPa at 50 °C for 15 min; (f) the 300 MPa + 50 oC group was treated under 300 MPa at 50 °C for 15 min.

Line 81: der: whats does this mean?

Response: thanks for your suggestion! We changed it accordingly.

Line 81: "the pressure of 300 MPa for 15 min". I did not understand this part of the sentence.

Response: thanks for your suggestion! We changed it accordingly.

Line 85: Change to stored at 4 °C until further analysis.

Response: thanks for your suggestion! We changed it accordingly.

Line 86: It was previously mentioned that 6 groups were vacuum-packed. Please clarify all information in this section. I suggest that the authors present a table or figure of the experimental design, as this section is too confusing to understand the treatments.

Response: thanks for your suggestion! We changed it accordingly.

The treatments used (all in the same day) were: (a) The control group (0.1 MPa) was no-pressurized (NP) at room temperatures (RT); (b) the 150 MPa group was treated under 150 MPa in a high-pressure equipment (Stansted Fluid Power Ltd., Harlow, England) at RT for 15 min; (c) the 300 MPa group was treated under 300 MPa at RT for 15 min; (d) the 0.1 MPa+50 oC group was treated under NP at 50 °C for 15 min; (e) the 150 MPa + 50 oC group was treated under 150 MPa at 50 °C for 15 min; (f) the 300 MPa + 50 oC group was treated under 300 MPa at 50 °C for 15 min.

Line 99: Authors should not use the parameter gumminess. Chewiness only applies to solid products, whereas gumminess only applies to semi-solid products. Gumminess is mutually exclusive with chewiness since a product would not be both semi-solid and solid at the same time.

Response: thanks for your suggestion! In our samples, the control collagen tissue was solid products. After treating under high pressure and heating, the collagen tissue became loose, formed to a gel and changed to a semi-solid products. Based on the special materials, we determined both gumminess and chewiness parameters.

Line 104: I suggest removing this sentence since the authors are explained below how each one is calculated.

Response: thanks for your suggestion! We deleted it accordingly.

Line 106: How many measurements for each treatment?

Response: thanks for your suggestion! We changed it accordingly.

All measurements were taken at room temperature and repeated 10 times.

Line 109: In this type of citation, the name of the authors should be added. Ex: by Lou et al. [14].

Response: thanks for your suggestion! We changed it accordingly.

Line 123: "eosinThe". Please, add a space.

Response: thanks for your suggestion! We changed it accordingly.

Line 131: In this type of citation, the name of the authors should be added. Ex: by Lou et al. [14].

Response: thanks for your suggestion! We changed it accordingly.

Lines 146-147: The author used respectively but mentioned just one method. Please, rewrite.

Response: thanks for your suggestion! We changed it accordingly.

Lines 158-159: Was this wavelength used to determine the four analyses? (cross-linked degree, total collagen, DCN, and GAGs)? This was not clear.

Response: Yes, this wavelength can be used to determine the 4 analyses. The HRP conjugated rabbit anti-mouse secondary antibodies has a special absorption at 450 nm. Four different analyses used 4 different primary antibodies but the same secondary HRP conjugated antibodies.

Line 163: At the beginning of the material and methods, it was mentioned that each group had 10 samples and in the statistical analysis, it was informed that the results were expressed as means of fifteen replicate samples. Please clarify.

Response: It's our clerical error. We have checked and corrected it.

Results

Line 171: In the Material and Methods, it was indicated that the temperature of these treatments was 25 ºC and here, the temperature was not mentioned. Please add the information of the temperature throughout the manuscript.

Response: thanks for your suggestion! We changed the expression as room temperature accordingly.

Line 173: Same as previously mentioned about the gumminess determination.

Response: thanks for your suggestion! We changed it accordingly.

The treatments used (all in the same day) were: (a) The control group (0.1 MPa) was no-pressurized (NP) at room temperatures (RT); (b) the 150 MPa group was treated under 150 MPa in a high-pressure equipment (Stansted Fluid Power Ltd., Harlow, England) at RT for 15 min; (c) the 300 MPa group was treated under 300 MPa at RT for 15 min; (d) the 0.1 MPa+50 oC group was treated under NP at 50 °C for 15 min; (e) the 150 MPa + 50 oC group was treated under 150 MPa at 50 °C for 15 min; (f) the 300 MPa + 50 oC group was treated under 300 MPa at 50 °C for 15 min.

Line 249: "onlyreduced". Please, add a space

Response: thanks for your suggestion! We changed it accordingly.

Line 264: In this type of citation, the name of the authors should be added. Ex: by Lou et al. [14].

Response: thanks for your suggestion! We changed it accordingly.

Lines 307-308: Rewrite to make it clearer.  

Response: thanks for your suggestion! We changed it to “Therefore, the reduction of cross-linking could contribute to the destruction of thermal stability of collagen”.

Conclusions

The conclusion is basically presenting the results again. The conclusion is an answer to your goals, not a summary of results. I suggest the authors rewrite this section.

Response: thanks for your suggestion! We changed it accordingly.

The single high pressure treatment failed to improve the hardness and chewiness of pig trotter with weak denaturation of collagen proteins. The high pressure/heating combination at 300 MPa + 50 °C improved the textural parameters pig trotter by inducing the ultra-structural collapse of collagen fibers, reducing cross-links, DCN and GAGs levels and unfolding triple-helix structure of collagen in pig trotter.

 Line 325: "wereprocessed" Please, add a space.

Response: thanks for your suggestion! We changed it accordingly.

Round 2

Reviewer 1 Report

The manuscript has been sufficiently improved.

Reviewer 2 Report

The authors have considered all my comments